# Measurement of the Growth of the Main Commercial Rays (*Raja clavata*, *Raja brachyura*, *Torpedo marmorata*, *Dipturus oxyrinchus*) in European Waters Using Intercalibration Methods

**DOI:** 10.3390/biology13010020

**Published:** 2023-12-29

**Authors:** Andrea Bellodi, Pierluigi Carbonara, Kirsteen M. MacKenzie, Blondine Agus, Karen Bekaert, Eleanor S. I. Greenway, Maria C. Follesa, Manfredi Madia, Andrea Massaro, Michele Palmisano, Chiara Romano, Mauro Sinopoli, Francesca Ferragut-Perello, Kélig Mahé

**Affiliations:** 1Stazione Zoologica Anton Dohrn, Contrada Porticatello 29, 98167 Messina, Italy; andrea.bellodi@szn.it (A.B.); or blondine.agus@unica.it (B.A.); 2Department of Life and Environmental Sciences, University of Cagliari, 09126 Cagliari, Italy; follesac@unica.it (M.C.F.); m.madia@studenti.unica.it (M.M.); 3Fondazione COISPA ETS, Via dei Trulli 18/20, 70126 Bari, Italy; carbonara@coispa.it (P.C.); palmisano@coispa.eu (M.P.); 4Ifremer, Fisheries Laboratory, Channel and North Sea Fisheries Research Unit, 62200 Boulogne-sur-Mer, France; kirsteen.mackenzie@ifremer.fr; 5ILVO—Flanders Research Institute for Agriculture, Fisheries and Food, 8400 Oostende, Belgium; karen.bekaert@ilvo.vlaanderen.be; 6Aquaculture and Fisheries Group, Wageningen University and Research, 6708 PB Wageningen, The Netherlands; eleanor.greenway@wur.nl; 7SZN—Sicily Marine Centre, Lungomare Cristoforo Colombo 4521, 90149 Palermo, Italy; mauro.sinopoli@szn.it; 8APLYSIA—Ricerche Applicate all’Ecologia e alla Biologia Marina, 57128 Livorno, Italy; andrea.massaro@aplysia.it; 9Department of Biology, Ecology and Earth Science, University of Calabria, Via Pietro Bucci, 87036 Arcavacata di Rende, Italy; chiara.romano@unical.it; 10Centre Oceanogràfic de les Balears (COB-IEO), CSIC, Moll de Ponent s/n, 07015 Palma, Spain; francesca.ferragut@ieo.csic.es

**Keywords:** growth, precision, accuracy, batoids

## Abstract

**Simple Summary:**

Poor quality of biological information, such as age and growth parameters, could be a source of variability with a significant impact on stock assessment results. Concerning the ageing process, variability is frequently linked to differences in the interpretation of calcified structures. The evaluation of precision and accuracy therefore represents a keystone in the ageing procedure.

**Abstract:**

The intercalibration of age readings represents a crucial step in the ageing procedure; the use of different sampling methods, structures, preparation techniques, and ageing criteria can significantly affect age and growth data. This study evaluated the precision and accuracy of ageing for the most important North Atlantic (NA) and Mediterranean (M) ray species, *Raja clavata*, *Raja brachyura*, *Torpedo marmorata*, and *Dipturus oxyrinchus*, through exchange exercises carried out by readers from different laboratories. In addition, growth parameters were estimated from the obtained data. A total of 663 individual batoids were analysed. *R. clavata* and *R. brachyura* samples were obtained from both the NA and the M, while vertebral centra of *T. marmorata* and *D. oxyrinchus* were only available for the M. High reading variability was observed for all four evaluated species in terms of CV, APE, and PA. *D. oxyrinchus* and *T. marmorata* showed relatively slow growth and the von Bertalanffy model with fixed *t*_0_ and Gompertz’s model were, respectively, the most precise models for each of these species. In *R. brachyura*, females had a faster growth rate compared to combined sexes. The vbt0p proved the most precise model for describing growth in this species, and no statistical differences were found between the NO and the M. For *R. clavata*, the best-fitting model was the vbt0p for females and males in the NO and for females from the M, while the best-fitting model for males from the M and sexes combined for both areas was log.p. Distinct growth patterns were observed between the two study areas.

## 1. Introduction

Elasmobranchs are globally indicated as one of the most highly threatened vertebrate groups [1], due to their life history traits, which are typical of k-selected life strategy species [2]. These traits make these fish highly susceptible to anthropogenic impacts and in particular to fishing pressure [1,3]. Several shark and skate species are commercially valuable for their fins, meat, liver oil, gill rakers, or leather and are an important food resource [4]. Sharks and rays were once considered a lower-value bycatch of more profitable fisheries stocks, such as tuna, cod, and shrimps [5]. The rising demand for marine products, coupled with the decline of valuable target stocks has, however, resulted in rising catches and retention of these taxa [6]. Sharks and skates are found as bycatch species in fisheries worldwide, and the Mediterranean and Atlantic Ocean are areas with a high level of bycatch for both sharks and rays at a global scale [7]. Fishing pressure thus puts cartilaginous fish at a higher risk, which emphasises the need for effective conservation and management measures. Shark and ray fisheries have only recently been subject to management strategies, and attention to their need for conservation has grown [8]. The International Union for the Conservation of Nature’s (IUCN) Red List of Threatened Species estimates that about a quarter of elasmobranch species are threatened with extinction (i.e., assessed or estimated to be Vulnerable, Endangered, or Critically Endangered), and overfishing is the principal threat behind elasmobranch population declines [1]. Despite the advances in shark and ray fisheries management, there are concerns that elasmobranch fisheries are following the pattern of unregulated fisheries, resulting in wild population decline and collapse of stocks [9]. Indeed, shark and ray landings increased from 1950 (the first year of data collection) to the peak year at the beginning of the 2000′s, then subsequently declined in the following years [10].

Conservation and fisheries management measures require a solid assessment of population status, which must be based on reliable information regarding species life history traits. Age and growth data are essential to obtain mortality data and productivity estimations, which are crucial in stock assessments [11]. Describing growth parameters has been historically more difficult in cartilaginous fish than bony fish [12]. Ageing in bony fish involves the use of calcified structures, such as otoliths, which are absent in elasmobranch species. Cartilaginous structures in elasmobranchs, such as vertebrae, have low calcification levels and often require additional enhancements of growth band visibility with the use of staining techniques. These characteristics make ageing of elasmobranchs a rather complex process [13].

Inaccurate population assessments resulting in stock collapse have been caused in some cases by poor-quality ageing data [14]. A growing body of research has therefore been conducted to increase the accuracy of age data, particularly within the European Union Data Collection Framework [15]. Variations in age data between institutes may occur due to sampling methods (e.g., commercial fishing or scientific surveys) [16], the use of different structures (vertebral centra, spines, scales, etc.), preparation techniques [17], and the ageing criteria used [11,18]. Variable levels of fishing pressure [19,20] and spatial variations related to environmental conditions or genetic factors [21,22] could be the basis of variation in growth patterns in fish stocks, including contiguous populations. The fitting of growth models to age data could also be a source of variability with significant impact on stock assessment results. Finally, the level of reader experience can be a very important additional source of variability [23]. All these factors can compromise both precision and accuracy of age data and consequently the analysis of the level of population exploitation. Unreliable scientific advice can result from using incorrect growth parameters or age-at-length keys to translate size distribution into age structure. If the age of a population is overestimated, the stock assessment will incorrectly predict that fishing mortality will be lower because the population will be composed of older specimens. Conversely, if the age is underestimated, fishing mortality will be overestimated, and the population will appear to be composed of younger specimens [24]. Additionally, age and growth have an impact on how data on natural mortality and maturity at age are estimated. As a result, these measures also impact how recruitment strength and biomass of spawning stock are estimated. Ultimately, the most significant impact is related to short-term stock status forecasts and the corresponding management actions [18].

In this context, an age calibration exercise was initiated on the most important North Atlantic and Mediterranean ray species from commercial (*Raja clavata; Raja brachyura*) [10,25] and conservation (*Torpedo marmorata*; *Dipturus oxyrinchus*) [26,27] perspectives. The exercise aimed to assess the precision of age and growth data for these species. The present study, resulting from the exercise, aims to provide reliable age and growth data for *R. clavata*, *R. brachyura*, *T. marmorata* and *D. oxyrinchus* in European waters (North Atlantic Ocean and Mediterranean Sea). These data will then contribute to more effective management plans. As some recent papers stated that alternative growth models could offer a better fit for cartilaginous fish age-at-length data, in particular for batoids (e.g., [28,29]), we applied a number of alternative growth models to age-at-length data in addition to the frequently used von Bertalanffy function.

## 2. Materials and Methods

### 2.1. Sampling

Individuals were sampled in two areas in the North Atlantic Ocean (North Sea, ICES area 27.4; Eastern Channel, ICES area 27.7.d) and in two areas in the Mediterranean Sea (Ligurian and North Tyrrhenian Seas, GSA 9; western part of Sardinia, GSA 11.1) between 2010 and 2020 from commercial sampling and during scientific surveys. All individuals were taken to the laboratory for accurate measurements. Each individual was measured to the nearest mm for total length (TL) and to the nearest g for total weight (WT). Finally, the sex of each specimen was recorded.

### 2.2. Ageing Procedures

Vertebrae were excised from the spine during dissection and subsequently used to estimate age data. The preparation methods used varied between institutions, where vertebrae sampled from the North Atlantic Ocean were stained using varying concentrations of crystal violet and read as whole structures, while vertebral samples from the Mediterranean Sea were sagittally sectioned and left unstained (Figure 1). A number of vertebrae were collected for each individual. One vertebra per individual was photographed using a binocular microscope under transmitted light. 

Alternating translucent and opaque bands were visible in the vertebra of these elasmobranchs. It was assumed that each annual growth ring consisted of one opaque and one translucent band, as is standard in temperate fish sclerochronological analyses. The age was therefore expressed in consistent age groups, e.g., a fish in age group 0 lived between 1 day and 364 days (i.e., between hatching and before 1 year), as recommended by international expert groups [30,31,32].

### 2.3. Ageing Data Precision

To limit interpretation error and reading bias, each individual was analysed by eleven European readers from Italy, Greece, Belgium, the Netherlands, and France during the European exchange in 2022 to evaluate precision. Precision is defined as the reproducibility of repeated measurements on a given scale, whether or not measurements are accurate [30]. Precision was measured from the average percent error (APE), the percentage agreement (PA), and the coefficient of variation (CV). The formula presented by Beamish and Fournier [33] was used to calculate APE:(1)$$APEj\left(\%\right)=100\frac{1}{R}\sum_{i=1}^{R}\frac{\left|Xij+Xj\right|}{Xj}$$
where *Xij* is the *i*th age determination of the *j*th fish, *Xj* is the average age calculated for the *j*th fish, and *R* is the number of times each fish was aged. CV and PA within one year (+/−1 yr) were proposed by [24]: $$PA=\frac{\sum \left|n_{diff}\leq 1\right|}{n}$$
$$CVj\left(\%\right)=100\cdot \frac{\sqrt{\sum_{i=1}^{R}\frac{{\left(X_{ij}-X_{j}\right)}^{2}}{R-1}}}{x_{j}}$$
where *R* is the number of times each fish is aged, *Xij* the *i*(th) age determination of the *j*(th) fish, *X_j_* is the mean age calculated for the *j*(th) fish, and *n_diff_* is the difference in age determination between the readings of two readers.

### 2.4. Growth Model Estimation

Non-linear growth models were fitted to length-at-age data. Mean body growth patterns of the commercial ray species sampled were described using four different growth models including the following:
the unconstrained von Bertalanffy model [34] (vbp):
$${TL}_{t}={TL}_{\infty }\cdot (1-e^{-K\left(t-t_{0}\right)})$$
the von Bertalanffy model with forced *t*_0_ = 0 (vbt0p):
$${TL}_{t}={TL}_{\infty }-({TL}_{\infty \;}e^{-Kt})$$
the Gompertz model [35] (vbL1p):
$${TL}_{t}={TL}_{\infty }\cdot e^{lnln\;\left(\frac{{TL}_{1}}{{TL}_{\infty }}\right)\;e^{-K(t-1)}}$$
the logistic model [36] (log.p):
$${TL}_{t}=\frac{{TL}_{\infty }}{1+\left(\left(\frac{{TL}_{\infty }}{{TL}_{1}}\right)-1\right)*e^{-Kt}}$$
where ${TL}_{1},{TL}_{t},\mathrm{and}{TL}_{\infty }$ are, respectively, the length at age 1, at age *t* and the asymptotic length, and *K* is the rate at which the asymptote is reached, also called the growth coefficient. 

### 2.5. Data Analysis

For each individual, the total length and the age group were estimated according to the sex and/or geographical sampling area. With these all-individual data, all growth models were tested and the best growth model was identified as the one that minimised the small sample bias-corrected form of the Akaike Information Criterion (AICc) [37,38]. The AICc balances the trade off between the quality of fit and the number of parameters used [39] while accounting for small sample bias, and is defined as follows:$$AICc=2k-2ln\;(TL)+\frac{2k(k+1)}{n-k-1}$$
where *n* is the sample size, *k* is the total number of parameters of the model, and *TL* is its likelihood.

Fish growth was estimated using the growth performance index (φ) [40]:(2)$$\varphi =log\;K+2\;log\;({TL}_{\infty })$$

Growth performance index was more appropriate for growth comparison versus comparison of *TL*_∞_ and *K* individually, as these two parameters are highly correlated [41].

The lifespan (*t_max_*) was estimated from the empirical relationship with growth rate k [42] as follows:$$t_{max}=-\frac{ln\;(1-0.95)}{k}$$

The Chen test [43] was used to look for potential differences in growth between areas.

## 3. Results

### 3.1. Sample Composition

A total of 663 individual batoids were analysed in the present study. Table 1 reports the specific data of the sample composition. *R. clavata* and *R. brachyura* samples were obtained both from the Atlantic Ocean and the Mediterranean Sea, while vertebral centra of *T. marmorata* and *D. oxyrinchus* were only available for the Mediterranean basin. *R. clavata* was the most sampled species, with 224 females (131–955 mm TL) and 204 males (209–900 mm TL) (Table 1), followed by *R. brachyura* for which a total of 115 samples were collected (54 females 175–990 mm TL; 61 males 70–955 mm TL). A total of 60 specimens of *T. marmorata* and 61 specimens of *D. oxyrinchus* were analysed from the Mediterranean basin. TLs of these species ranged between 127 and 557 mm and between 220 and 1120 mm, respectively, for sexes combined (Table 1).

### 3.2. Ageing Precision

The ageing precision evaluation returned relatively high reading variability for all four evaluated species. CV and APE values ranged between 47 and 49% and between 33 and 37%, respectively, for *R. brachyura* and *T. marmorata*. For *D. oxyrinchus* and *R. clavata*, slightly more precise results were observed (CV 30–34%; APE 21–26%) (Table 2). Nonetheless, the PA was similar for all evaluated species (44–52%) (Table 2). Figure 2 shows the CV and PA values, specifically for each modal age of the four examined species against the standard deviation. In general, PA is observed to be higher in younger age classes, showing values close to 75% (Figure 2D, *D. oxyrinchus*) or higher (Figure 2C, *T. marmorata*), while in the two species belonging to the genus *Raja* this value seems more stable around the 50% in all modal ages, with an inflection observed for specimens older than 5–6 years (Figure 2A,B). CV values also show this trend where lower reader variation is observed for younger modal ages and this variation increases with modal age (Figure 2). Logically, the reading standard deviation seemed to follow the opposite tendency, with higher values for older modal ages (Figure 2).

### 3.3. Growth Parameters

Due to the relatively small sample number of *D. oxyrinchus* and *T. marmorata*, it was only possible to estimate growth for the sexes combined. The AICc indicated that the most precise models were the von Bertalanffy model with fixed *t*_0_ and Gompertz’s model in fitting the observed data for *D. oxyrinchus* and *T. marmorata*, respectively (Table 3). Both species appeared to grow relatively slowly (*D. oxyrinchus* k = 0.101; *T. marmorata* k = 0.175) and to be capable of a long lifespan, with estimations of up to 30 years for *D. oxyrinchus* and up to 17 for *T. marmorata* (Table 3).

The age-at-length data were also insufficient to model the growth of male *R. brachyura*, thus the species growth pattern was estimated only for females and combined sexes both in the Mediterranean Sea and the Atlantic Ocean (Table 3). In both study areas, the vbt0p was the most precise model in describing the species growth for sexes combined according to the AICc, while for females this was the vbp model (Table 3). When considering combined sexes, *R. brachyura* appears to be a relatively slow growing species, however females showed a faster growth rate (k = 0.397 in Atlantic Ocean, k = 0.429 in Mediterranean Sea). The observed ages ranged between 0 and 8 years in the Atlantic Ocean and between 0 and 10 years in the Mediterranean Sea. This resulted in a higher estimated lifespan for this species in the Mediterranean Sea (18 years) compared to in Atlantic waters (12 years) (Table 3). The Chen test comparison of the obtained vbp growth curve did not show a statistical difference between the areas (Fobs < Fcrit), thus this species follows a similar growth pattern in both the Atlantic and Mediterranean regions (Figure 3; Table 3).

Finally, the vbt0p was the best-fitting model to the observed age-at-length data for *R. clavata* females and males in the Atlantic Ocean and for females from Mediterranean Sea. The logistic model (log.p) returned the best-fitting results (AICc) for both males from Mediterranean Sea and both sexes combined for both the investigated areas (Table 3). *R. clavata* seems to be a relatively slow-growing species, with males that appear to be capable of growing faster yet show a shorter lifespan than females. In contrast to *R. brachyura*, two distinct growth patterns were observed for *R. clavata* between the Atlantic Ocean and the Mediterranean Sea (Chen test, Fobs > Fcrit). The Atlantic Ocean population produced higher growth rates compared to the Mediterranean population (Figure 3). 

The age-at-length data for all species are plotted in Figure 4 with the growth model that provided the best-fitting results for the observations. 

## 4. Discussion

This paper presents the first attempt at an intercalibration of age readers for elasmobranch species at a European level. For the first time, eleven international readers from five European countries took part in an age reading exercise that involved over 600 calcified structures extracted from four different batoid species. Vertebral preparation methods varied between institutions, although only vertebral sections were available for *T. marmorata* and *D. oxyrinchus*. Both vertebral sections and whole vertebrae were obtained for *R. brachyura* and *R. clavata*, which were collected in the Mediterranean Sea and the North-eastern Atlantic Ocean, respectively. Vertebrae were therefore analysed with the same preparation method for each geographical sampling area.

In consideration of the large number of scientists involved, the results obtained from the analysis of the ageing precision and reproducibility, although appearing far from the thresholds usually considered acceptable for elasmobranch ageing studies (*sensu* [12]), can be considered encouraging. Indexes such as the CV, the APE, and particularly the PA can easily be negatively affected by a high number of readers. Additionally, readers from different countries, while having a good experience level in interpreting hard structures, were also asked to read structures prepared following different protocols from those to which they were accustomed. This difference potentially played a role in the age reading variability observed in this study. 

It is well known that reader experience is the most important factor affecting ageing precision. This has been confirmed in other age calibration studies, which compared other potential sources of bias such as the identification of first annulus or the interpretation of possible false rings [23]. The present study also seems to confirm this assumption as, despite the application of different structure preparation methods, the best outcomes in terms of reading precision were obtained for *R. clavata*. This species is analysed in all the laboratories involved and ageing was consequently more familiar to the readers. The intercalibration results obtained should therefore be considered encouraging, and future reading exercises and workshops must be endorsed.

Although the growth modelling of the analysed species was not the principal purpose of the age reading exchange, the comparison of the obtained growth parameters with those reported in the extant literature (Table 4) revealed, in most cases, no major differences. The growth patterns of the two species investigated in both the Atlantic and the Mediterranean Sea, *R. clavata* and *R. brachyura*, appeared similar to previous observations in the two areas (Table 4). The main differences may be ascribed to the growth model selected, as only the common von Bertalanffy function was considered in many studies, or to the age estimation method (e.g., tagging [44]) and the hard structure used (e.g., caudal thorns [45,46]). The logistic, the vbp, and the vbt0p models provided the best fit to the length-at-age data following model selection with AIC. This is in accordance with the study of Thys et al. [29] where the best-fitting models were the logistic and vbp model (vbt0p not tested). Growth patterns for *T. marmorata* appeared to be in line with the literature [47,48] indicating this batoid as a slow-growing and long-lived species. Conversely, the calculated growth parameters for *D. oxyrinchus*, while comparable to those in Sardinian [49] and Tunisian waters [50], appeared rather different from those estimated by Yigin and Ismen [51]. The data from the present study indicated a much faster growth rate and an asymptotic length of almost half the size previously reported. 

It is also interesting to note that the different growth rates determined in the analysed species seem unlikely to be related to their trophic level. In fact, the two species that showed the faster growth rates, *R. brachyura* and *R. clavata*, are recognized as a specialist bony fish predator and as a generalist feeder, respectively [52]. Similarly for the two slow-growing species, *T. marmorata* specializes in hunting fish [48] and *D. oxyrinchus* is more generalist [51]. Nonetheless, it could be interesting to investigate further how interactions with the environment could affect the growth of these species, and future studies on this aspect should be endorsed and welcomed.

The present study also observed different growth patterns for *R. clavata* and *R. brachyura* caught in the Atlantic Ocean and the Mediterranean Sea. *R. brachyura* produced similar growth patterns for both investigated areas, while *R. clavata* appeared to be capable of growing faster and larger in the Atlantic Ocean compared to the Mediterranean Sea. Although different preparation methods were used between the different areas for both species, i.e. whole vertebrae from the Atlantic and sectioned vertebrae from the Mediterranean, differences in growth patterns between areas were only observed for one species, *R. clavata*. This therefore suggests that these divergences in growth are not caused by the preparation method but may be linked to other factors such as different environmental conditions (e.g., water temperature, prey availability, nutrient levels, pollution, etc.) [21,53], fishing pressures [54], or strong regional genetic differentiation between Atlantic and Mediterranean populations of *R. clavata* [55]. This is the first study investigating growth of these skates and including samples collected from the two different areas, and represents a first step towards a better understanding of the factors that may influence growth patterns of these batoids.

**Table 4 biology-13-00020-t004:** Biogeographic comparison of the biological parameters of the investigated elasmobranch species with sampling details (number of samples, observed maximum TL and Age), and details of the growth model (vbp = unconstrained von Bertalanffy growth model, vbt0p = von Bertalanffy model with forced *t*_0_ = 0, vbL1p = Gompertz’ growth model, log.p = logistic growth model), with the type of data and the parameters of the growth model and growth performance index (ϕ).

Species	Sector	Geographical Area	Sampling				Growth Model						Lifespan (Years *t_max_*)	ϕ	Sources
			Sex	N	TL (mm) Max	Age (Years) Max	Vertebrae Preparation/Ageing Method	Growth Model	*TL* _∞_	k	*t* _0_	*TL* _1_			
*Raja brachyura*	**Atlantic Ocean**	**North-east Atlantic**	**F + M**	**45**	**990**	**8**	**whole**	**vbt0p**	**1052.19**	**0.25**	**−0.72**	**-**	**12**	5.43	**this study**
	Atlantic Ocean	Irish seas	F		910		whole	vbp	1443.00	0.19	−0.31			5.60	[56]
			M		777				1194.00	0.26	−0.31			5.57	
	Atlantic Ocean	Irish seas	F	141	1080		sectioned		1547.00	0.13	−0.84			5.49	[57]
			M	127	1090				1458.00	0.15	−0.93			5.49	
	Atlantic Ocean	Irish seas	F		1120		tagging	vbt	1184.00	0.19	−0.80			5.43	[44]
			M		1150				1150.00	0.19	−0.18			5.40	
	Atlantic Ocean	Portugal waters	F + M	139	1106		caudal thorns	vbt	133.50	0.12	0.29			3.33	[45]
	Atlantic Ocean	North-east Atlantic	F	31	911		whole	log	1020	0.24	-				[29]
			M	25	876			vbp	857	0.18					
	**Mediterranean Sea**	**Sardinian seas**	**F + M**	**60**	**955**	**10**	**sectioned**	**vbt0p**	**1166.32**	**0.17**	**−1.33**	**-**	**18**	**5.36**	**this study**
	Mediterranean Sea	Sardinian seas	**F + M**	168	955	16	sectioned	vbt	1113.4	0.1	−1.19	**-**		5.09	[58]
*R. clavata*	**Atlantic Ocean**	**North-east Atlantic**	**F + M**	**214**	**955**	**10**	**whole**	**log.p**	**897.25**	**0.46**	**-**	**229.03**	**23**	**5.57**	**this study**
	Atlantic Ocean	Irish seas	F	93			sectioned	vbt	1395.00	0.09	−1.74			5.26	[57]
			M	165					1065.00	0.14	−1.74			5.19	
	Atlantic Ocean	Welsh seas	F	135			whole	vbt	1176.00	0.16	−0.71			5.34	[59]
			M	54					1009.00	0.18	−0.95			5.26	
	Atlantic Ocean	Portuguese waters	F + M	251	913	10	caudal thorns	vbt	1280.00	0.12	−0.61			5.28	[46]
	Atlantic Ocean	North-East Atlantic	F	45	906		whole	log	831	0.354	-				[29]
			M	42	785			vbp	807	0.17					
	**Mediterranean Sea**	**Central-western Mediterranean**	**F + M**	**212**	**819**	**11**	**sectioned**	**log.p**	**713.14**	**0.49**	**-**	**237.48**	**15**	**5.40**	**this study**
	Mediterranean Sea	South Adriatic Sea	F + M	291	890	12	sectioned	vbt	986	0.18	−0.95			5.24	[60]
	Mediterranean Sea	Northern Tyrrhenian Sea	F + M	262	800		sectioned	log.p	709.2	0.55		190		5.44	[61]
	Mediterranean Sea	Central Tyrrhenian Sea	F + M	118	864			vbt	929.6	0.21	−0.73			5.26	
	Mediterranean Sea	Sardinian seas	F + M	235	824			vbt	876.1	0.14	−1.79			5.03	
	Mediterranean Sea	Western Ionian Sea	F + M	**105**	**826**			vbt	870.50	0.19	−0.88			5.16	
	Mediterranean Sea	Tunisian seas	F	160	1040	15	sectioned	vbt	1146.00	0.11	−1.23			5.16	[62]
			M	125	850	12			1008.00	0.14	−1.13			5.15	
	Mediterranean Sea	Strait of Sicily	F	224			sectioned	vbt	1265.00	0.10	−0.51			5.20	[63]
			M	200					1162.00	0.11	−0.41			5.16	
*Dipturus oxyrinchus*	**Mediterranean Sea**	**Sardinian seas**	**F + M**	**61**	**1120**	**13**	**sectioned**	**vbt0p**	**1461.87**	**0.10**	**−1.90**		**30**	**5.33**	**this study**
	Mediterranean Sea	Sardinian seas	F + M	130	1155	17	sectioned	vbL1p	1275.5	0.14				5.36	[49]
	Mediterranean Sea	Turkish seas	F + M	169	885	9	sectioned	vbt	2564.6	0.04	−1.17			5.42	[51]
	Mediterranean Sea	Tunisian seas	F	175	1050	25	sectioned	vbt	1239	0.08	−1.26		38	5.09	[50]
			M	110	950	26			1021	0.12	−1.18		26	5.10	
*Torpedo marmorata*	**Mediterranean Sea**	**Sardinian seas**	**F + M**	**60**	**557**	**10**	**sectioned**	**vbL1p**	**581.72**	**0.18**	**-**	**214.56**	**17**	**4.78**	**this study**
	Mediterranean Sea	Sardinian seas	F	65	560	17	sectioned	vbL1p	622.44	0.16			23	4.78	[48]
			M	77	432	10		vbt	485.01	0.14			15	4.52	
	Mediterranean Sea	Turkish Sea	F + M	117	560	6	sectioned	vbt	573.17	0.19	−0.39			4.79	[47]

## 5. Conclusions

This study was the result of an international exchange with eleven readers representing five European countries. Age and growth parameters were successfully evaluated for four batoid species, *R. clavata*, *R. brachyura*, *T. marmorata*, and *D. oxyrinchus*, sampled in European waters, namely the Atlantic Ocean and the Mediterranean Sea. Although the precision of the age readings in this study is relatively low, the results are still encouraging considering the large number of age readers. Precision was higher at lower ages and decreased with age for all species. Alternative growth models were used to describe the age-at-length data where different models performed better than others, depending on the species, sex, and location. 

The outcomes of this research, while preliminary, emphasise the need for intercalibration events involving large numbers of different laboratories and scientists from different countries. In this way, it is possible to increase ageing data quality for these ecologically important species, while providing solid inputs for their stock evaluation and management.

## Figures and Tables

**Figure 1 biology-13-00020-f001:**
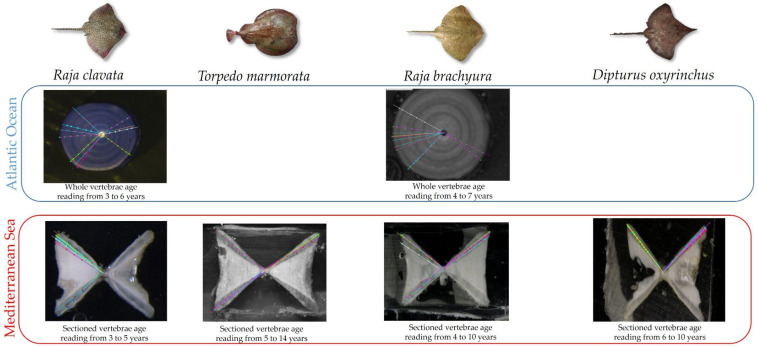
Summary of the preparation methods used for the analysed species in the Atlantic Ocean and Mediterranean Sea. Dots and lines represent the structure interpretation by different readers.

**Figure 2 biology-13-00020-f002:**
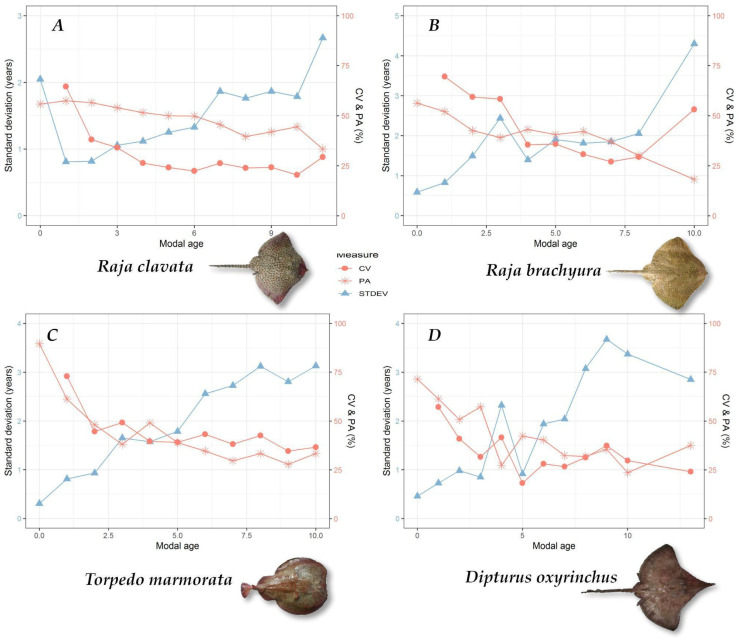
Coefficient of variation (CV), percentage of agreement (PA), and standard deviation values for each modal age of *R. clavata* (**A**), *R. brachyura* (**B**), *T. marmorata* (**C**), and *D. oxyrinchus* (**D**).

**Figure 3 biology-13-00020-f003:**
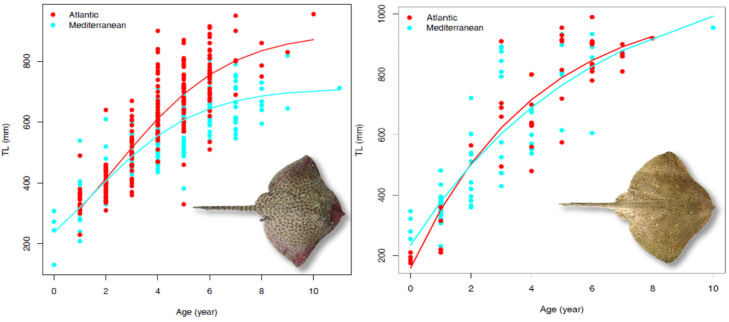
Comparison of the estimated growth curves and the observed age-at-length data obtained for *R. clavata* (**left**) and *R. brachyura* (**right**) in the Atlantic Ocean and Mediterranean Sea.

**Figure 4 biology-13-00020-f004:**
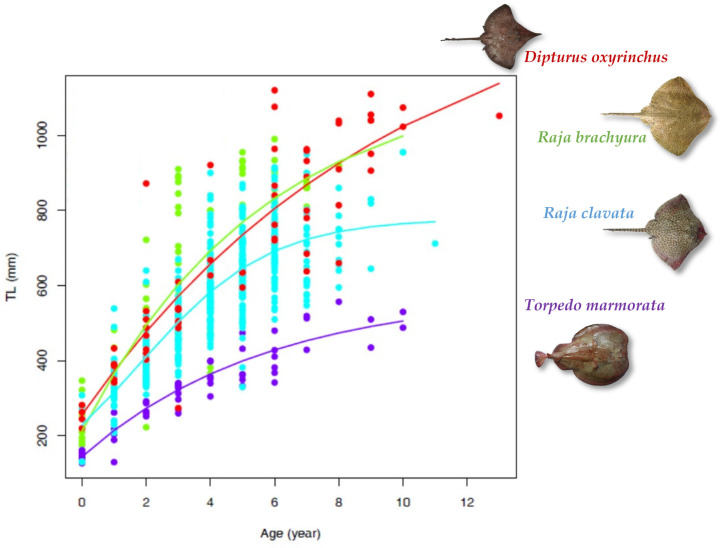
Age-at-length data for each species with the estimated growth curves from each best-fitting model: von Bertalanffy with constrained *t*_0_ (vbt0p) curves for *R. brachyura* and *D. oxyrinchus*, logistic curve (log.p) for *R. clavata*, and Gompertz’s curve (vbT1p) for *T. marmorata*.

**Table 1 biology-13-00020-t001:** Sample composition per species with number of specimens (N), mean TL with standard error (SE), size range (in mm), mean age (in years) with SE, and age range for females (F), males (M), and sex combined (F + M).

Species	Sex	N	TL (mm)		Age (Year)	
			Mean ± SE	Range	Mean ± SE	Range
*Dipturus oxyrinchus*	F + M	61	697.2 ± 33.5	220–1120	5 ± 1.28	0–13
	F	41	714.6 ± 44.8	220–1120	5 ± 1.71	0–13
	M	20	661.5 ± 43.4	245–963	4 ± 1.34	0–9
*Raja brachyura*	F + M	115	585.2 ± 22.6	70–990	3 ± 0.46	0–10
	F	54	574.9 ± 21.8	175–990	3 ± 0.68	0–8
	M	61	594.3 ± 28.7	70–955	3 ± 0.64	0–10
*R. clavata*	F + M	428	574.6 ± 7.18	131–955	4 ± 0.14	0–11
	F	224	587.0 ± 10.7	131–955	4 ± 0.27	0–11
	M	204	561.0 ± 9.29	209–900	4 ± 0.21	0–9
*Torpedo marmorata*	F + M	60	321.7 ± 14.9	127–557	4 ± 0.90	0–10
	F	29	368.8 ± 23.8	138–557	5 ± 1.67	0–10
	M	31	277.7 ± 14.5	127–432	2 ± 0.54	0–6

**Table 2 biology-13-00020-t002:** Summary of the ageing precision results, with vertebral centra preparation method, for each elasmobranch investigated species.

Species	N Vertebrae	Preparation Method		N Readers	Ageing Precision Results		
		Whole	Sectioned		CV	PA	APE
*Raja brachyura*	115	45	70	11	49%	44%	37%
*R. clavata*	428	215	213	11	30%	52%	21%
*Dipturus oxyrinchus*	61	-	61	9	34%	45%	26%
*Torpedo marmorata*	60	-	60	9	47%	49%	33%

**Table 3 biology-13-00020-t003:** Summary of the growth modelling results obtained for each species for females (F), males (M), and sexes combined (F + M) in the different geographical areas investigated. N is the number of analysed specimens, with information on size composition (TL max and TL min, in mm) and observed age range (age max and age min, in years). Growth model indicates the best-fitting model to the observed age-at-length data according to the AICc (vbp = unconstrained von Bertalanffy growth model, vbt0p = von Bertalanffy model with forced *t*_0_ = 0, vbL1p = Gompertz’ growth model, log.p = logistic growth model); *TL*_∞_ is the asymptotic length (mm); k is the growth coefficient; TL1 and *t*_0_ are, respectively, the length at age 1 and the theoretical length at time 0. The estimation of the lifespan and the growth performance index (ϕ) are also reported.

Species	Area	Sex	Sampling					Growth						
			N	TL Max	TL Min	Age Min	Age Max	Growth Model	*TL*_∞_ (mm)	k	*t* _0_	*TL* _1_	Lifespan (*t_max_*)	φ
*Raja brachyura*	Atlantic Ocean	F + M	45	990	191.25	0	8	vbt0p	1052.196	0.245	−0.664		12	5.433
		F	24	910	220	1	7	vbp	911.278	0.397	0.000		8	5.519
		M	**-**	**-**	**-**	**-**	**-**	**-**	**-**	**-**	**-**	**-**	**-**	**-**
	Mediterranean Sea	F + M	60	955	301	0	10	vbt0p	1166.318	0.168	−1.331		18	5.360
		F	37	955	347	0	10	vbp	885.650	0.429	0.000		7	5.527
		M	**-**	**-**	**-**	**-**	**-**	**-**	**-**	**-**	**-**	**-**	**-**	**-**
*Dipturus oxyrinchus*	Mediterranean Sea	F + M	61	1120	252.5	0	13	vbt0p	1461.872	0.101	−1.904		30	5.335
*R. clavata*	Atlantic Ocean	F + M	214	955	339.444	1	10	log.p	897.251	0.458		229.03	7	5.567
		F	126	955	341.25	1	10	vbt0p	1267.266	0.132	−1.127		23	5.325
		M	88	900	335.833	1	9	vbt0p	901.265	0.231	−0.701		13	5.273
	Mediterranean Sea	F + M	212	819	239.25	0	11	log.p	713.144	0.490		237.49	6	5.396
		F	97	819	188	0	11	vbt0p	858.364	0.196	−1.250		15	5.160
		M	115	790	290.5	0	9	log.p	674.832	0.537		244.40	6	5.388
*Torpedo marmorata*	Mediterranean Sea	F + M	60	557	142.88	0	10	vbL1p	581.715	0.175		214.56	17	4.772

## Data Availability

Data are contained within the article and in the report of the Workshop on age reading and maturity stages of elasmobranch species (WKARMSE 2023).

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
