# Peer review of "Measurement of the Growth of the Main Commercial Rays (*Raja clavata*, *Raja brachyura*, *Torpedo marmorata*, *Dipturus oxyrinchus*) in European Waters Using Intercalibration Methods"

_biology, 2023, doi:10.3390/biology13010020_

Round 1
Reviewer 1 Report
Comments and Suggestions for Authors
I think this is a very interesting and informative study on the growth rates of some common and less common batoids. I think however that it could be improved by a few relatively minor changes:
There is no discussion of the choice of taxa. Clearly R. clavata is commercially very important whilst some of the other species are in a concerning state. However there is a noticeable omission of smaller presumably rapid growing species (of Raja and Leucoraja spp) that are also commercially important and appear to be more resistant to fishing pressure. Is that because the were not available?
One very important character that was not really discussed was sexual maturity. Of course assessing this in females is not easy unless there are developing eggs, but in males this is easy die to clasper development. Conservation efforts need to take this into account, and know at what size and age the species mature. This is especially important the comparing otherwise similar Raja and Dipturus, in which the latter appears more vulnerable to fishing.
Some discussion of the ecological position of the species would be useful. R. clavata is (as far as I am aware) a generalist with a very diverse diet and wide depth range; the other species are more dietary specialised (to teleosts) and may have a lesser depth range. Does that impact on the results?
The different growth rates in R. clavata in the two seaways ia very interesting. Why may this be the case? Could this be temperature or nutrient levels? If the latter, nutrients would be higher in the Atlantic than the Med, but also higher in the western than eastern Med. (essentially related to river input of dissolved material).
Author Response
Dear reviewer 1
Rev#1
I think this is a very interesting and informative study on the growth rates of some common and less common batoids. I think however that it could be improved by a few relatively minor changes:
Answer: We are grateful for the appreciation of our work and for the effort that the reviewer made for improve the quality of our MS
There is no discussion of the choice of taxa. Clearly R. clavata is commercially very important whilst some of the other species are in a concerning state. However, there is a noticeable omission of smaller presumably rapid growing species (of Raja and Leucoraja spp) that are also commercially important and appear to be more resistant to fishing pressure. Is that because the were not available?
Answer: Yes, we agree that it could have been interesting including also other species and, particularly, some fast-growing batoids. However, the species were selected to take advantage of those that were analysed in the European cartilaginous fish age reading exchange organized by ICES in 2022 and the subsequent workshop held in June 2023 (ICES Workshop on Ageing and Maturity in Elasmobranch Species). In this regard, the species were chosen because the laboratories involved had shared the samples. Moreover, considering that the main aim of the project was to achieve a good intercalibration between Atlantic and Mediterranean labs, we tried to avoid species only present in one of these regions (such as endemics) in order to work on species that would be familiar to every reader involved.
One very important character that was not really discussed was sexual maturity. Of course assessing this in females is not easy unless there are developing eggs, but in males this is easy die to clasper development. Conservation efforts need to take this into account, and know at what size and age the species mature. This is especially important the comparing otherwise similar Raja and Dipturus, in which the latter appears more vulnerable to fishing.
Answer: Yes we agree, all the reproductive features are mandatory information for the correct management of any fishery resource since they are important inputs of stock assessment routines. However, we choose to not cover these aspects because the main aim of our manuscript, taking advantages of the above-mentioned age calibration exercise, was to provide for the first time in a wide European area an accurate evaluation of the ageing precision and reproducibility on elasmobranchs, comparing also different preparation techniques. In this regard, we aim to provide a first solid step for future studies regarding elasmobranchs’ growth that can successively be used, together with reproductive data, for the correct management of this species.
Some discussion of the ecological position of the species would be useful. R. clavata is (as far as I am aware) a generalist with a very diverse diet and wide depth range; the other species are more dietary specialised (to teleosts) and may have a lesser depth range. Does that impact on the results?
Answer: Thanks for the suggestion. A small section dealing with this aspect has been implemented in the discussion (L. 327-337)
The different growth rates in R. clavata in the two seaways ia very interesting. Why may this be the case? Could this be temperature or nutrient levels? If the latter, nutrients would be higher in the Atlantic than the Med, but also higher in the western than eastern Med. (essentially related to river input of dissolved material).
Answer: It is indeed interesting. In the last part of the discussion, we listed some of the possible causes (which now includes also “nutrient level” as suggested). Unfortunately, our data does not allow us to investigate deeply the cause of these discrepancies, as that would require a wide environmental study. But, as far as our paper is concerned, the key result was to rule out the possibility that these differences were linked to the different methods used in the laboratories involved.
thank a lot for your work
the authors

Reviewer 2 Report
Comments and Suggestions for Authors
PLS See attachment

Please confirm that you appreciate to be notified with the final status of the paper
Author Response
Dear reviewer 2
General comments:
This MS by Bellodi et al. evaluated precision and accuracy of ageing for the most important North Atlantic (NA) and Mediterranean (M) ray species, Raja clavata, Raja brachyura, Torpedo marmorata, and Dipturus oxyrinchus, through exchange exercises carried out by readers from different laboratories. The result suggests that D. oxyrinchus and T. marmorata showed relatively slow growth. In R. brachyura, females had a faster growth rate compared to combined sexes. The vbt0p proved the most precise model for describing growth in this species, and no statistical differences were found between NO and M. For R. clavata, the best fitting model was the vbt0p for females and males in the NO and for females from M, while the best fitting model for males from M and sexes combined for both areas was log.p. Distinct growth patterns were observed between the two study areas.
Overall, I thought this was a well-executed study in a system with limited previous knowledge of this level on this specific topic (age and growth). I appreciated their multi-faceted approach and the time course involved in this work and think it add significant merit to their work. I do think that their overall conclusions were pretty general and not particularly unexpected or insightful as they were presented in the discussion. Additionally, there were some methodological parts I was confused about and some aspects of the results that were either missing or under-explored.
Overall, I think this is good work, but should undergo some revision before acceptance. I will outline my concerns below:
Answer: We are grateful to the reviewer for the work done on our manuscript. We truly believe that their efforts have contributed to increase the overall MS quality.
(1)
Introduction: I think there is too many references, reach 49. Even after each sentence, there are a few references. The author should have some of their own original viewpoints and ideas, not always quote others, even every sentence. In addition, the author should cite some classic references. More references are not necessarily better.
Answer: We recognize the reviewer's argument. We cut the number of references in this section to 29 because there were too many of them. This approach was chosen since, as a paper introduction should give enough context for the subject matter and indicate why the manuscript may be significant, we may have gone overboard in this regard. We now think that this part as a whole seems more linear.
(2)
Materials and Methods
Line 123 Format issues
Answer: Ok, corrected (L. 122)
Line 130 The author wrote “The gonads were observed macroscopically todetermine the sex and sexual maturity stage”, but I cannot find the sexual maturity stage in the results, only determine the sex (male and female). Please explain this problem.
Answer: True, the sentence has been changed (L. 129)
Line 133 “Vertebrae were excised from the spine during dissection and subsequently used to estimate age data.” I think if estimate age data, using the otolith is more accurate, why the author chose Vertebrae?
Answer: Unfortunately, otoliths are absent in cartilaginous fish, this fact often makes the ageing of these species a more challenging process than in bony fish as we stated in lines 79-83 of introduction. Nonetheless, the vertebral centra usually provide solid results and this type of analysis has been validated for some species through both direct and indirect methods.
(3)
Results
3.1. Morphological parameters, there is no describe information on age, only TL?
Answer: The age is discussed in the subsequent paragraphs, however we do agree that this paragraph title was confusing so we changed to “Sample composition” (L. 195)
Table 1, Mean and SD Separation is unscientific. The data should express as Mean±±SD, Please use Scientific notation.
Answer: Ok, corrected (L. 210)
Table 1, Age, Mean and SD, the data should express Mean±±SD,
Answer: Ok, corrected (L. 210)
Table 1, the same species appears for the first time used for the full name, and abbreviations are used for subsequent occurrences. Other tables (2,3,4) also need to be modified. Raja brachyuran is OK, Raja clavata change to R. clavata.
Answer: Ok, corrected
Why SD is much greater than Mean? Are some ages negative?
Answer: No, there were of course no negative ages. The standard deviation is a predictive index of the dispersion of values around the mean. In this regard it is sensible to outliers that are most common in a study like ours where numerous laboratories and readers are involved, each characterized by different degrees of experience or used to different preparation techniques. This might have caused such variation, consequently, understanding the cause of this variation while trying to reduce it is precisely the reason behind the promotion of intercalibration exercises similar to the one that originated our study. Nonetheless, we recognize such index could be confusing in situations similar to ours, for this reason we changed SD to the standard error, so that results might be easier to interpret.
3.2. Ageing precision, there is no describe information on Fig. 2A and B? only Fig. 2C and D.
Answer: True, thanks for noticing it. Information added (L. 218-222)
Line 262, “in” should Lowercase letters
Answer: Ok, corrected (L. 263)
(4)
Discussion
Table 4 compared the results in this study and other studies. It still introduce the result. I suggest Table 4 should move the result section. Generally, tables do not appear in Discussion better, only in result.
Answer: Although some of our results are present (but none is reported here for the first time), the table’s aim is to provide a clear comparison of all the data available in the literature for the species examined that are analysed and discussed in the Discussion. In this regard, considering that such table mostly shows results from other studies that are discussed here, we’d prefer to keep the table in this section, as often happens in other papers.
Line 295, One more space.
Answer: Ok, corrected (L. 291)
Line 346, Table 4, Bellodi et al., 2021 should change to [].
Answer: Ok, corrected (Tab. 4)
Check the references again.
Answer: the reference list has been updated.
thank a lot for your review
best regards
the authors

Round 2
Reviewer 2 Report
Comments and Suggestions for Authors
I think the author carefully revised Ms according to the reviewer's suggestions.
I agree to accept after minor revision
In 2. Materials and Methods part
Add a new paragraph
2.5 Data analysis explain how the author processing raw data.
This is necessary for publish.
Comments on the Quality of English Language
Minor editing of English language required
Author Response
Dear Reviewer 2,
we added the new chapter :
"
2.5. Data analysis
For each individual, the total length and the age group were estimated according to the sex and/or geographical sampling area. With these all individual data, all growth models were tested and the best growth model was identified as the one that minimised the small sample bias-corrected form of the Akaike Information Criterion (AICc) [37,38]. The AICc balances the trade-off between the quality of fit and the number of parameters used [39] while accounting for small sample bias, and is defined as:
where n is the sample size, k is the total number of parameters of the model and TL is its likelihood.
Fish growth was estimated using the growth performance index (φ) [40]:
φ = log K + 2 log
Growth performance index was more appropriate for growth comparison versus comparison of ??∞ and K individually, as these two parameters are highly correlated [41].
The lifespan (tmax) was estimated from the empirical relationship with growth rate k [42] as follows:
The Chen test [43] was used to look for potential differences in growth between areas."
Best regards
the authors
